



Atmospheric
Chemistry
and Physics

# Relating geostationary satellite measurements of aerosol optical depth (AOD) over East Asia to fine particulate matter (PM$_{2.5}$): insights from the KORUS-AQ aircraft campaign and GEOS-Chem model simulations

Shixian Zhai[1], Daniel J. Jacob[1], Jared F. Brewer[1], Ke Li[1], Jonathan M. Moch[1], Jhoon Kim[2,3], Seoyoung Lee[2], Hyunkwang Lim[2], Hyun Chul Lee[3], Su Keun Kuk[3], Rokjin J. Park[4], Jaein I. Jeong[4], Xuan Wang[5], Pengfei Liu[6], Gan Luo[7], Fangqun Yu[7], Jun Meng[8], Randall V. Martin[9], Katherine R. Travis[10], Johnathan W. Hair[10], Bruce E. Anderson[10], Jack E. Dibb[11], Jose L. Jimenez[12], Pedro Campuzano-Jost[12], Benjamin A. Nault[12,a], Jung-Hun Woo[13], Younha Kim[14], Qiang Zhang[15], and Hong Liao[16]

[1]Harvard John A. Paulson School of Engineering and Applied Sciences, Harvard University, Cambridge, MA, USA
[2]Department of Atmospheric Sciences, Yonsei University, Seoul, Republic of Korea
[3]Samsung Particulate Matter Research Institute, Samsung Advanced Institute of Technology, 130 Samsung-ro, Yeongtong-gu, Suwon-si, Gyeonggi-do, Republic of Korea
[4]School of Earth and Environmental Sciences, Seoul National University, Seoul, Republic of Korea
[5]School of Energy and Environment, City University of Hong Kong, Hong Kong SAR, China
[6]School of Earth and Atmospheric Sciences, Georgia Institute of Technology, Atlanta, GA, USA
[7]Atmospheric Sciences Research Center, University at Albany, Albany, New York, USA
[8]Department of Atmospheric and Oceanic Sciences, University of California, Los Angeles, California, USA
[9]Department of Energy, Environmental and Chemical Engineering, Washington University in St. Louis, St. Louis, MO, USA
[10]NASA Langley Research Center, Hampton, VA, USA
[11]Institute for the Study of Earth, Oceans, and Space, University of New Hampshire, Durham, NH, USA
[12]Department of Chemistry, Cooperative Institute for Research in Environmental Sciences, University of Colorado, Boulder, CO, USA
[13]Department of Civil and Environmental Engineering, Konkuk University, Seoul, Republic of Korea
[14]International Institute for Applied Systems Analysis (IIASA), 2361 Laxenburg, Austria
[15]Department of Earth System Science, Tsinghua University, Beijing, China
[16]Jiangsu Key Laboratory of Atmospheric Environment Monitoring and Pollution Control, Collaborative Innovation Center of Atmospheric Environment and Equipment Technology, School of Environmental Science and Engineering, Nanjing University of Information Science and Technology, Nanjing, China
[a]now at: Center for Aerosol and Cloud Chemistry, Aerodyne Research, Inc., Billerica, MA, USA

Correspondence: Shixian Zhai (zhaisx@g.harvard.edu)

Received: 17 May 2021 – Discussion started: 27 July 2021
Revised: 16 October 2021 – Accepted: 19 October 2021 – Published:

**Abstract.** Geostationary satellite measurements of aerosol optical depth (AOD) over East Asia from the Geostationary Ocean Color Imager (GOCI) and Advanced Himawari Imager (AHI) instruments can augment surface monitoring of fine particulate matter ($PM_{2.5}$) air quality, but this requires better understanding of the AOD–$PM_{2.5}$ relationship. Here we use the GEOS-Chem chemical transport model to analyze the critical variables determining the AOD–$PM_{2.5}$ relationship over East Asia by simulation of observations from satellite, aircraft, and ground-based datasets. This includes the detailed vertical aerosol profiling over South Korea from the KORUS-AQ aircraft campaign (May–June 2016) with concurrent ground-based $PM_{2.5}$ composition, $PM_{10}$, and AERONET AOD measurements. The KORUS-AQ data show that 550 nm AOD is mainly contributed by sulfate–nitrate–ammonium (SNA) and organic aerosols in the planetary boundary layer (PBL), despite large dust concentrations in the free troposphere, reflecting the optically effective size and high hygroscopicity of the PBL aerosols. We updated SNA and organic aerosol size distributions in GEOS-Chem to represent aerosol optical properties over East Asia by using in situ measurements of particle size distributions from KORUS-AQ. We find that SNA and organic aerosols over East Asia have larger size (number median radius of 0.11 μm with geometric standard deviation of 1.4) and 20 % larger mass extinction efficiency as compared to aerosols over North America (default setting in GEOS-Chem). Although GEOS-Chem is successful in reproducing the KORUS-AQ vertical profiles of aerosol mass, its ability to link AOD to $PM_{2.5}$ is limited by under-accounting of coarse PM and by a large overestimate of nighttime $PM_{2.5}$ nitrate. The GOCI–AHI AOD data over East Asia in different seasons show agreement with AERONET AODs and a spatial distribution consistent with surface $PM_{2.5}$ network data. The AOD observations over North China show a summer maximum and winter minimum, opposite in phase to surface $PM_{2.5}$. This is due to low PBL depths compounded by high residential coal emissions in winter and high relative humidity (RH) in summer. Seasonality of AOD and $PM_{2.5}$ over South Korea is much weaker, reflecting weaker variation in PBL depth and lack of residential coal emissions.

## 1   Introduction

$PM_{2.5}$ (particulate matter with aerodynamic diameter less than 2.5 μm) in surface air is a severe public health concern in East Asia, but surface monitoring networks are too sparse to thoroughly assess population exposure. Satellite observations of aerosol optical depth (AOD) can provide a valuable complement (van Donkelaar et al., 2015). Geostationary satellite sensors, including the Geostationary Ocean Color Imager (GOCI) launched by the Korea Aerospace Research Institute (KARI) in 2011 (M. Choi et al., 2016, 2018,

2019) and the Advanced Himawari Imager (AHI) launched by the Japanese Meteorological Agency (JMA) in 2014 (Lim et al., 2018, 2021), offer the potential for high-density mapping of $PM_{2.5}$ over East Asia (Chen et al., 2019; Wei et al., 2021a). However, more confidence is needed in relating AOD to $PM_{2.5}$. Here we evaluate the capability of the GEOS-Chem chemical transport model (CTM) to simulate AOD–$PM_{2.5}$ relationships over East Asia, exploiting in situ aircraft measurements of vertical aerosol profiles and optical properties from the joint NASA–NIER Korea–United States Air Quality (KORUS-AQ) field study in May–June 2016 (Crawford et al., 2021; Peterson et al., 2019; Jordan et al., 2020) together with GOCI–AHI geostationary satellite data and surface measurement networks. This enables us to identify critical variables and uncertainties for inferring $PM_{2.5}$ from satellite AOD data.

A number of past studies have used satellite AOD data to infer surface $PM_{2.5}$ using physical and statistical models. The standard geophysical approach has been to use a CTM, such as GEOS-Chem, to compute the $PM_{2.5}$ / AOD ratio (Liu et al., 2004; van Donkelaar et al., 2006, 2015; Xu et al., 2015; Geng et al., 2017), with recent applications correcting for CTM biases using available $PM_{2.5}$ surface network data (Brauer et al., 2016; van Donkelaar et al., 2016, 2019; Hammer et al., 2020). An alternative approach is to use artificial intelligence algorithms to relate satellite AOD to $PM_{2.5}$ by training on the surface network data (Hu et al., 2017; Chen et al., 2018; Xiao et al., 2018; Wei et al., 2021a; Wei et al., 2021b; Pendergrass et al., 2021) and sometimes including CTM values as predictors (Di et al., 2019; Xue et al., 2019). Yet another approach is to assimilate the satellite-measured AODs in a CTM and correct in this manner the $PM_{2.5}$ simulation, although this requires attribution of model AOD errors to specific model parameters (Kumar et al., 2019; Saide et al., 2014; Sekiyama et al., 2010; Cheng et al., 2019). In all of these approaches, a better physical understanding of the AOD–$PM_{2.5}$ relationship as simulated by CTMs can greatly enhance the capability to infer $PM_{2.5}$ from AOD data.

AOD measures aerosol extinction (scattering and absorption) integrated over the atmospheric column so that its relationship to 24 h average surface $PM_{2.5}$ (the standard air quality metric) depends on the aerosol vertical distribution and optical properties, ambient relative humidity (RH), diurnal variation in $PM_{2.5}$, and contribution from coarse particulate matter to AOD. Airborne measurements of aerosol vertical profiles (without species information) in East Asia are limited (Zhang et al., 2006; Liu et al., 2009; Zhang et al., 2009; Sun et al., 2013; Li et al., 2017), and speciated vertical profiles are rarer. AOD is highly sensitive to RH (Brock et al., 2016; Latimer and Martin et al., 2019; Saide et al., 2020), but the impact from RH uncertainty on AOD simulation lacks evaluation. In addition, because the AOD is a daytime measurement that needs to be related to 24 h average $PM_{2.5}$, the diurnal variation in $PM_{2.5}$ needs to be understood (Guo et al., 2017; Lennartson et al., 2018). Finally, although there

Atmos. Chem. Phys., 21, 1–17, 2021                                                                    https://doi.org/10.5194/acp-21-1-2021

**Table 1.** Surface site observations used in this work (2016).

| Variable | Number of sites |
| --- | --- |
| $PM_{2.5}$ in East China[a] | 598 |
| $PM_{2.5}$ in South Korea[b] | 130 |
| $PM_{2.5}$ composition in South Korea (May–June 2016)[c] | 7 |
| AERONET total and fine-mode AOD in East China[d] | 5 |
| AERONET total and fine-mode AOD in South Korea[d] | 10–21[e] |

[a] Hourly $PM_{2.5}$ from the China National Environmental Monitoring Centre (CNEMC; http://www.quotsoft.net/air/, last access: 12 March 2021) in East China, including only sites with more than 90 % data coverage in each month of 2016. Quality control of the CNEMC dataset is described in our previous study (Zhai et al., 2019). The $PM_{2.5}$ measurements are made at reference RH $\leq 35$ %. [b] Hourly $PM_{2.5}$ from the AirKorea network (https://www.airkorea.or.kr, last access: 20 April 2021), with the same data selection criteria as for East China. The $PM_{2.5}$ measurements are made at reference RH $\leq 35$ %. [c] Major $PM_{2.5}$ components including sulfate, nitrate, ammonium, organic carbon, and black carbon at seven supersites in South Korea during KORUS-AQ (May–June 2016; J. Choi et al., 2019). The mass concentration of organic carbon is converted to that of organic aerosol with a multiplicative factor of 1.8 based on KORUS-AQ observations (Kim et al., 2018). [d] AODs are from the AERONET Version 3 Level 2.0 all-points database (https://aeronet.gsfc.nasa.gov/, last access: 23 July 2021), except that AODs at the Xuzhou site in East China are from the Version 3 Level 1.5 database. AOD at 500 nm ($AOD_{500\,nm}$) is converted to 550 nm ($AOD_{550\,nm}$) using the Ångström exponent at 500 nm ($AE_{500\,nm}$) following

$AOD_{550\,nm} = AOD_{500\,nm} \left( \frac{550}{500} \right)^{-AE_{500\,nm}}$. [e] AERONET AODs in South Korea are from 10 sites for the full year of 2016 and 21 sites during KORUS-AQ.

**Table 2.** KORUS-AQ aircraft observations used in this work (May–June 2016).

| Variable | Instrument |
| --- | --- |
| Aerosol extinction profile at 532 nm | HSRL[a] |
| Aerosol scattering coefficient at 550 nm | TSI nephelometers[b] |
| Aerosol absorption coefficient at 532 nm | PSAPs[c] |
| Aerosol dry size distribution | TSI LAS[d] |
| Bulk aerosol ionic composition | SAGA[e] |
| Sub-micron non-refractory aerosol composition | HR-ToF-AMS[f] |
| Black carbon concentration | HDSP2[g] |
| Relative humidity | DLH[h] |

[a] NASA Langley airborne high-spectral-resolution lidar (HSRL) (Hair et al., 2008; Scarino et al., 2014). [b] NASA Langley TSI-3563 nephelometers (Ziemba et al., 2013). [c] Radiance Research three-wavelength particle soot absorption photometers (PSAPs; Ziemba et al., 2013). [d] In situ particle size distributions over the 0.1–5.0 μm diameter range from the TSI laser aerosol spectrometer (LAS) Model 3340. [e] University of New Hampshire (UNH) Soluble Acidic Gases and Aerosol (SAGA) instrument (Dibb et al., 2003). The cutoff aerodynamic diameter of the inlet is around 4 μm, corresponding to a geometric particle diameter of 2.5 μm (McNaughton et al., 2007, 2009). [f] University of Colorado Boulder high-resolution time-of-flight aerosol mass spectrometer (HR-ToF-AMS; DeCarlo et al., 2006; Nault et al., 2018; Guo et al., 2021). [g] NOAA humidified dual single particle soot photometer (HDSP2; Lamb et al., 2018). [h] NASA diode laser hygrometer (DLH; Podolske et al., 2003).

are studies on the optical depth of coarse-mode desert dust (Eck et al., 2010; Ridley et al., 2016), there has been to our knowledge no study of how coarse anthropogenic PM may contribute to the AOD measurements. Coarse anthropogenic PM (distinct from desert dust) is known to be high over East Asia (Chen et al., 2015; Dai et al., 2018).

## 2 Data and methods

### 2.1 Observations

We use observations over China and South Korea from multiple platforms including surface sites, aircraft, and satellites (Tables 1 and 2). Surface data (Table 1) include $PM_{2.5}$ from national observation networks in China (Zhai et al., 2019) and South Korea (Jordan et al., 2020), speciated $PM_{2.5}$ at seven supersites in South Korea during KORUS-AQ (J. Choi et al., 2019), and ground-based AODs from the AERONET network at 5 sites in East China and 10 sites in South Korea (21 sites during KORUS-AQ). We use total and fine-mode AODs at 500 nm wavelength from the AERONET Version 3 spectral deconvolution algorithm (SDA) Version 4.1 Retrieval Level 2.0 database (Giles et al., 2019; O'Neill et al., 2003). The AERONET AODs at 500 nm are converted to 550 nm using total and fine-mode Ångström exponents at 500 nm for consistency with the satellite AOD data.

The KORUS-AQ campaign (Table 2) includes 20 flights over the Korean peninsula and the surrounding ocean from 2 May to 10 June 2016, with vertical profiling up to 8 km altitude. We use the aircraft observations of remote and in situ aerosol extinction (scattering + absorption) coefficients, dry aerosol number size distributions, sub-micron non-refractory aerosol composition, bulk aerosol ionic composition, black carbon (BC), and relative humidity (RH).

Geostationary satellite AOD at 550 nm is retrieved by the Yonsei aerosol retrieval (YAER) algorithm for the GOCI (Choi et al., 2016, 2018) and AHI (Lim et al., 2018) instruments, with GOCI covering East China and South Korea and AHI covering the broad East Asia region. AOD from GOCI and AHI has a 6 km × 6 km spatial resolution and 2.5 min (AHI) to 1 h (GOCI) temporal resolution for 8 h per day (09:30 to 16:30 local time). We use the fused AOD product generated from the Yonsei GOCI–AHI AOD retrievals, each using two different surface reflectance methods (Lim et al., 2021). Fusion of this four-member ensemble is done by the maximum likelihood estimate (MLE) method, with weighting and averaging based on errors determined by comparison to AERONET AOD. The fused satellite AOD product is shown by Lim et al. (2021) to have higher accuracy than its member products in comparison with AERONET data during the KORUS-AQ campaign. We refer to it as the "GEO satellite AOD" product in what follows.

### 2.2 GEOS-Chem simulation

We use GEOS-Chem version 12.7.1 (https://doi.org/10.5281/zenodo.3676008) in a nested-grid simulation at a horizontal resolution of $0.5° \times 0.625°$ over East Asia (100–145° E, 20–50° N). GEOS-Chem simulates detailed tropospheric oxidant–aerosol chemistry and is driven here by GEOS-FP-assimilated meteorological data from the NASA Global Modeling and Assimilation Office (GMAO). Boundary layer mixing uses the non-local scheme implemented by Lin and McElroy (2010). Dry deposition of gases and particles follows a standard resistance-in-series scheme (Zhang et al., 2001; Fairlie et al., 2007; Fisher et

al., 2011; Jaeglé et al., 2018). Wet deposition of gases and particles includes contributions from rainout, washout, and scavenging in convective updrafts (Liu et al., 2001; Amos et al., 2012; Q. Wang et al., 2011, 2014) with recent updates by Luo et al. (2019, 2020). We use pre-archived initial conditions from Zhai et al. (2021) and run the model from 1 December 2015 to 31 December 2016. The first month is used for spin-up, and the year 2016 is used for analysis.

GEOS-Chem has been used extensively to simulate $PM_{2.5}$ and its composition in East Asia (Geng et al., 2017; Li et al., 2016; J. Choi et al., 2019; Jeong et al., 2008; Park et al., 2021; Zhai et al., 2021). Here we use the bulk representation of aerosols including sulfate (Park et al., 2004; Alexander et al., 2009), nitrate (Jaeglé et al., 2018), primary and secondary organic aerosols (Pai et al., 2020), BC (Q. Wang et al., 2014), natural dust in four advected size ranges (Fairlie et al., 2007), anthropogenic fine dust (Philip et al., 2017), and sea salt in two size ranges (Jaeglé et al., 2011). Heterogeneous sulfate formation on aqueous aerosols is represented by a simplified parameterization scheme (Y. Wang et al., 2014), where the $SO_2$ uptake coefficient ($\gamma$) linearly increases from $1 \times 10^{-5}$ at RH $\leq 50\%$ to $2 \times 10^{-5}$ at RH $= 100\%$. The thermodynamic equilibrium of sulfate–nitrate–ammonium (SNA) aerosols with the gas phase is computed with ISORROPIA II (Fountoukis and Nenes, 2007; Pye et al., 2009), assuming an aqueous aerosol. We include reactive uptake on dust of acid gases ($HNO_3$, $SO_2$, and $H_2SO_4$), limited by consumption of dust alkalinity (Fairlie et al., 2010). The alkalinity of emitted dust is estimated by assuming 7.1 % $Ca^{2+}$ and 1.1 % $Mg^{2+}$ as alkaline cations by dust mass (Shah et al., 2020).

Monthly anthropogenic emissions are from the Multiresolution Emission Inventory in 2016 for China (MEIC; Zheng et al., 2018; http://meicmodel.org, last access: 10 November 2021) and from the KORUSv5 emission inventory in base year 2015 (Woo et al., 2020; http://aisl.konkuk.ac.kr/#/emission_data/korus-aq_emissions, last access: 10 November 2021) for other Asian countries and shipping emissions. MEIC over China applies weekly and diurnal scaling factors for all anthropogenic emissions (Zheng et al., 2018). The KORUSv5 agricultural $NH_3$ emissions apply the diurnal scaling factors from MEIC. Natural emissions include $NO_x$ from lightning (Murray et al., 2012) and soil (Hudman et al., 2012), MEGANv2 biogenic volatile organic compounds (VOCs) (Guenther et al., 2012), dust (Meng et al., 2021), and sea salt (Jaeglé et al., 2011). Open-fire emissions are from the Global Fire Emissions Database version 4 (GFED4; van der Werf et al., 2017).

## 2.3 AOD simulation

AOD in GEOS-Chem is diagnosed by integrating vertically the aerosol scattering and absorption coefficients obtained with a standard Mie calculation applied to assumed size distributions, hygroscopicity, refractive indices, and densities for individual aerosol components and summing over all components (Martin et al., 2003). Optical properties are listed in Table 3. Sulfate, nitrate, and ammonium share the same optical properties and are lumped as an SNA aerosol component for the purpose of optical calculations. All aerosol components except dust are assumed to follow lognormal size distributions. Dust includes seven size bins (centered at radii of 0.15, 0.25, 0.4, 0.8, 1.5, 2.5, and 4.0 μm) for optical calculations, with the smallest four bins partitioned by mass from the first advected dust bin ($< 2.5$ μm in geometric diameter) following Zhang et al. (2013). Dust particles follow a gamma size distribution within their optical size bins (Curci, 2012). The BC absorption enhancement from coating is as given by X. Wang et al. (2014).

Our initial simulations indicated that aerosol extinction coefficients from the standard GEOS-Chem version 12.7.1 underestimated in-situ-measured extinction coefficients during KORUS-AQ by 20 % on average (Fig. S1 in the Supplement). We traced this problem to bias in the assumed size distributions for SNA and organic aerosol, as shown in Sect. 3. Therefore, we recomputed the diagnostic AOD using updated lognormal size distributions for SNA and organic aerosol with number median radius $R_{N,med} = 0.11$ μm and geometric standard deviation $\sigma = 1.4$ based on KORUS-AQ observations instead of $R_{N,med} = 0.058$ μm and $\sigma = 1.6$ in the standard model version 12.7.1, which is derived from IMPROVE network measurements of aerosol mass scattering efficiency over North America (Latimer and Martin, 2019).

## 3 Aerosol concentrations and optical properties during KORUS-AQ

Here we use the KORUS-AQ aircraft observations and their simulation with GEOS-Chem to better understand the vertical distributions of different aerosol components contributing to AOD over South Korea. We begin with the mean vertical profile of aerosol mass and go on to examine the aerosol optical properties. This provides the basis for analyzing the observed vertical profile of aerosol extinction, its simulation by GEOS-Chem, and the consistency with GEO satellite and AERONET AOD measurements over South Korea during the KORUS-AQ period.

### 3.1 Vertical profile of aerosol mass

Figure 1 shows the mean aircraft vertical profiles of aerosol mass observed during KORUS-AQ and their simulation by GEOS-Chem. The KORUS-AQ aircraft sampled during the daytime, mainly between 09:00 and 15:00 local time. Here and elsewhere, the model is sampled along the flight tracks and at the flight times. The observed vertical distribution of aerosol mass concentrations (Fig. 1a) shows that 58 % of column aerosol mass is below 2 km altitude, which we define as the average planetary boundary layer (PBL) during KORUS-AQ, and 34 % is at 2–5 km altitude, which we define as the

**Table 3.** Aerosol optical properties[a]. Note that n/a stands for not applicable.

| Aerosol component | $R_{N,med}$, μm | $\sigma$ | Hygroscopicity[b] | Refractive index | $\rho$, g cm$^{-3}$ |
|---|---|---|---|---|---|
| SNA [c] | 0.11 | 1.4 | $\kappa = 0.61$ | $1.53 - 6.0 \times 10^{-3}i$ | 1.7 |
| Organic[c] | 0.11 | 1.4 | $\kappa = 0.1$ | $1.53 - 6.0 \times 10^{-3}i$ | 1.3 |
| BC | 0.020 | 1.6 | GADS | $1.75 - 4.4 \times 10^{-3}i$ | 1.8 |
| Sea salt (fine) | 0.085 | 1.5 | GADS | $1.5 - 1.0 \times 10^{-3}i$ | 2.2 |
| Sea salt (coarse) | 0.40 | 1.8 | GADS | $1.5 - 1.0 \times 10^{-3}i$ | 2.2 |
| Dust | Seven size bins | n/a | $\kappa = 0$[d] | $1.558 - 1.4 \times 10^{-3}i$ | 2.5–2.65[e] |

[a] Aerosol optical properties used in this work for computing aerosol scattering and absorption coefficients. Values are from the standard GEOS-Chem model version 12.7.1, except for the size distributions of SNA and organic aerosol, which are based on KORUS-AQ observations (see text). All aerosol components except dust have lognormal dry size distributions, where $R_{N,med}$ is the number median radius, and $\sigma$ is the geometric standard deviation. Refractive indices are for 550 nm wavelength; $\rho$ is the dry aerosol mass density.
[b] Hygroscopic growth for SNA and organic aerosol as a function of relative humidity (RH; %) is computed from $\kappa$-Kohler theory as a diameter growth factor $GF = (1 + \kappa \cdot RH/(100 - RH))^{1/3}$ (Latimer and Martin, 2019). Hygroscopic growth factors for other aerosol components are from the Global Aerosol Data Set (GADS) as tabulated in Chin et al. (2002) and Martin et al. (2003). [c] $R_{N,med}$ and $\sigma$ are fit to KORUS-AQ observations as described in the text. Standard GEOS-Chem v12.7.1 assumes $R_{N,med} = 0.058$ μm and $\sigma = 1.6$ (Latimer and Martin, 2019). [d] Hygroscopic growth of dust particles is assumed negligible. [e] Sub-micron dust particles have a density of 2.5 g cm$^{-3}$, while coarse-mode dust particles have a density of 2.65 g cm$^{-3}$. Dust size distribution is described in the text.

lower free troposphere (FT). The model has a similar vertical distribution (Fig. 1b), with 57 % of aerosol mass in the PBL and 36 % in the lower FT. SNA, organic aerosol, and dust each contribute about a third of aerosol mass in the PBL, while dust dominates in the lower FT both in the observations and in the model. The enhanced dust in the lower FT is driven by a few dust events, which the model reproduces (Fig. S2). Black carbon and sea salt (not shown) make only minor contributions to aerosol mass. The model underestimates sulfate by 28 % in the PBL, which leads to a 20 % overestimate of nitrate, with a canceling effect on the SNA mass simulation.

The GEOS-Chem simulation of organic aerosol in this work uses the simple scheme of Pai et al. (2020) and underestimates aircraft observations by 16 % in the PBL. Over 90 % of GEOS-Chem organic aerosol is secondary, consistent with observations (Fig. S4; Nault et al., 2018; Pai et al., 2020). GEOS-Chem simulation of the KORUS-AQ aerosol component profiles for different meteorological regimes is presented in Park et al. (2021).

### 3.2 Aerosol size distributions

Figure 2a shows the normalized dry aerosol number size distributions on each of the 20 flights and in 3 altitude bands: $< 1.5$, 3–5, and 6–7 km (60 lines). The spread in the size distributions above 1 μm in diameter reflects dust influence. We select measurements below 1.5 km altitude when SNA + organic aerosol mass concentrations are more than 4 times that of dust as defining the SNA + organic aerosol size distributions (green lines in Fig. 2a). Conditions dominated by SNA + organic aerosols define the lower envelopes of the ensemble of size distributions at diameter $> 1$ μm. SNA and organic aerosols were observed to have similar size distributions during KORUS-AQ (Kim et al., 2018).

Figure 2b converts the SNA + organic-aerosol-dominated number size distributions to volume size distributions. The

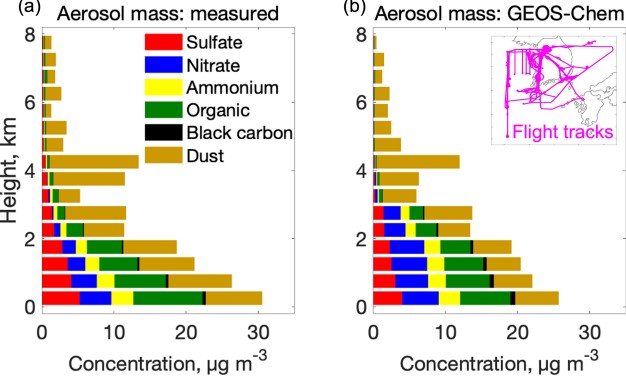

**Figure 1.** Vertical profiles of aerosol mass during KORUS-AQ. Panel **(a)** shows the mean vertical distributions of observed mass concentrations of major aerosol components at ambient temperature and pressure. Panel **(b)** is the same as **(a)** but from the GEOS-Chem model sampled along the flight tracks (inset). We derive dust concentration from SAGA $Ca^{2+}$ and $Na^+$ following Shah et al. (2020) by assuming that non-sea-salt $Ca^{2+}$ accounts for 7.1 % of dust mass: $[\text{dust}] = ([Ca^{2+}] - 0.0439 [Na^+]/2)/0.071$, where the brackets denote mass concentration. Modeled dust is shown for particles with geometric diameter $< 2.5$ μm, to be consistent with SAGA measurements (Table 2, footnote e). Measured sulfate, nitrate, ammonium, and organic aerosol concentrations are from the AMS instrument (values from the SAGA instrument are shown in Fig. S4). All data are averaged over 500 m vertical bins. Here and elsewhere, we excluded pollution plumes diagnosed by either $NO_2$ or $SO_2 > 10$ ppbv (3.4 % of all the data).

observed SNA + organic-aerosol-dominated size distribution is shifted toward larger sizes relative to the standard GEOS-Chem. The secondary maximum in the coarse mode could be due to dust. We fitted the observed SNA + organic aerosol size distributions to a lognormal distribution with volume median radius $R_{V,med} = 0.15$ μm and geometric standard

deviation $\sigma = 1.4$. The number median radius is derived from the volume median radius following Seinfeld and Pandis (2016):

$$\ln R_{N,med} = \ln R_{V,med} - 3\ln^2\sigma, \tag{1}$$

which yields $R_{N,med} = 0.11\,\mu m$. In comparison, the standard GEOS-Chem size distribution from Latimer and Martin (2019) has $R_{N,med} = 0.058\,\mu m$ and $\sigma = 1.6$. We adopt the observed lognormal size distribution parameters in what follows (Table 3).

## 3.3 Aerosol extinction and relation to AOD

Figure 3 shows the vertical profiles of ambient aerosol extinction coefficients and RH during KORUS-AQ. Vertical profiles of aerosol extinction were measured on the aircraft both remotely with the HSRL instrument (above and below the aircraft) and in situ with TSI-3563 nephelometers (for scattering) and PSAPs (for absorption). The two agree well, as shown in Fig. 3a. They indicate that 76 %–90 % of column aerosol extinction is in the PBL at 0–2 km altitude, and 9 %–19 % is in the lower FT at 2–5 km. Both measurements show that aerosol extinction is much more strongly weighted to the PBL than aerosol mass (Fig. 1).

Also shown in Fig. 3a are the contributions of individual aerosol components to the extinction profile, as computed from the GEOS-Chem optical properties (Table 3) applied to the observed mass concentrations. The sum shows a good match to the measured extinction coefficient profiles. The much larger contribution of the PBL to column aerosol extinction than to column mass is because aerosol mass in the lower FT is mainly composed of dust, whose mass extinction efficiency is much smaller than SNA and organic aerosols due to its coarse size and lack of hygroscopic growth (Fig. S5). The mean AOD inferred from the aircraft data is 0.36, and 59 % is contributed by SNA, 27 % by organic aerosol, 12 % by dust, and 2 % by BC. It is consistent with the mean AODs measured at AERONET stations in South Korea during KORUS-AQ (Fig. S6).

Figure 3b shows the GEOS-Chem simulation of aerosol extinction profiles for comparison to the observations in Fig. 3a. The model underestimates extinction coefficients by 20 % below 1 km altitude, leading to a 10 % underestimate of aircraft-inferred AOD, although there is no such underestimate in aerosol mass. This is caused by a negative RH bias in the GEOS-FP meteorological data used to drive GEOS-Chem, particularly under high-RH conditions (Fig. 3c), and is corrected if we apply the observed RH rather than the GEOS-FP RH to the GEOS-Chem aerosol mass concentrations (Fig. 3d).

## 4 AOD and surface particulate matter over South Korea during KORUS-AQ

Our analysis of Sect. 3 used the KORUS-AQ aircraft data together with GEOS-Chem to attribute AOD over South Korea to individual aerosol components and altitudes. We now take the next step of evaluating the capability of GEOS-Chem to independently simulate observed AODs and surface particulate matter concentrations.

Figure 4a shows the spatial distribution of the fused geostationary satellite (GOCI–AHI) AOD (GEO satellite AOD) during the KORUS-AQ period with AERONET total AOD added as circles. The GEO satellite AOD shows high values (0.5–0.6) along the west coast of South Korea, significantly correlated with AERONET total AOD with a spatial correlation coefficient ($R$) of 0.7. GEO satellite AOD is biased low at sites in the Seoul metropolitan area (SMA) and is biased high on the Yellow Sea islands, resulting in an overall $-10\,\%$ bias. The low biases in the SMA could be due to high-concentration aerosol pixels misidentified as clouds and/or possible issues with the aerosol type assumption in the aerosol retrieval, while the high biases on the Yellow Sea islands could result from uncertainties in the assumption of ocean surface reflectance, as has been discussed by Choi et al. (2016, 2018) and Lim et al. (2018, 2021). Sampling the AODs at or near the seven $PM_{2.5}$ supersites operating during KORUS-AQ shows no significant bias (inset values in Fig. 4a).

Figure 4b–e show the spatial distributions of GEOS-Chem AOD, surface $PM_{10}$ (particulate matter with aerodynamic diameter less than $10\,\mu m$), surface $PM_{2.5}$, and surface coarse PM ($PM_{10}$ minus $PM_{2.5}$; particulate matter with aerodynamic diameter less than $10\,\mu m$ and larger than $2.5\,\mu m$), with surface observations shown as circles and median values at the measurement sites inset. GEOS-Chem reproduces the satellite AOD enhancements along the west coast of South Korea, but the values are lower than observed, which we attribute to unaccounted coarse PM and negative RH bias as discussed below. Comparison of AERONET total and fine-mode AOD shows a 13 % contribution of coarse particles to total AOD. Comparison of GEOS-Chem to the fine-mode AERONET AOD, as shown in Fig. 4b, finds a 15 % underestimate that could be attributed to the negative bias in RH (Fig. 3c). Concurrent measurements of $PM_{10}$ and $PM_{2.5}$ at AirKorea sites show that coarse PM (median $21\,\mu g\,m^{-3}$) accounts for 41 % of total $PM_{10}$ ($50\,\mu g\,m^{-3}$), while coarse PM in GEOS-Chem is much lower ($3.5\,\mu g\,m^{-3}$; Fig. 4e). Therefore, about half of the GEOS-Chem underestimate of total AOD can be attributed to missing coarse PM, with the other half coming from negative RH bias. Coarse PM has a concentration larger than $10\,\mu g\,m^{-3}$ across South Korea, with higher concentration in the SMA ($\sim 30\,\mu g\,m^{-3}$) than in rural areas ($\sim 15\,\mu g\,m^{-3}$), implying an origin from both anthropogenic and natural sources (Fig. 4e).

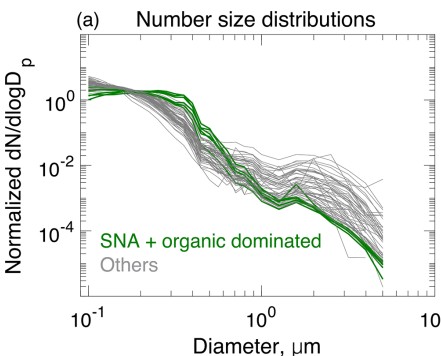
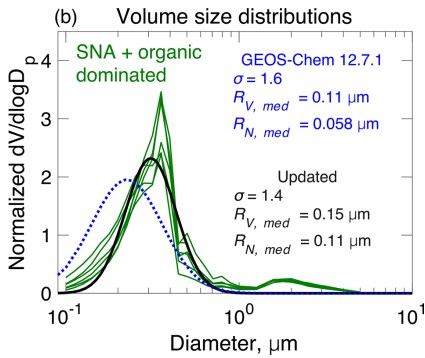

**Figure 2.** Aerosol dry size distributions measured in the KORUS-AQ aircraft campaign. Panel **(a)** shows mean normalized number size distributions measured on each of the 20 flights and for 3 altitude bins: < 1.5, 3–5, and 6–7 km (60 lines total). The SNA + organic-aerosol-dominated size distribution profiles are highlighted in color. Panel **(b)** shows normalized volume size distributions for conditions dominated by SNA + organic aerosols (green lines), along with a least-square fit to a lognormal distribution (black line) and the standard GEOS-Chem v12.7.1 size distribution from Latimer and Martin (2019) (dashed blue line). Normalization imposes an arbitrary value of unit area below each line. Lognormal distribution parameters are inset in panel **(b)**, including volume median radius ($R_{V,med}$), number median radius ($R_{N,med}$), and geometric standard deviation ($\sigma$).

GEOS-Chem overestimates surface $PM_{2.5}$ by 43 % over South Korea (Fig. 4d), in contrast to the simulation of AERONET fine-mode AOD (Fig. 4b). Figure 4f–j show the spatial distributions of major $PM_{2.5}$ components in GEOS-Chem (background) and measurements (filled circles). GEOS-Chem is not significantly biased relative to the observations for organic aerosol and BC and underestimates sulfate by 22 %. We find that the model bias for $PM_{2.5}$ is largely driven by nitrate, which is overestimated by a factor of 3 and leads to a 56 % overestimate of ammonium. By contrast, comparison to the KORUS-AQ data below 1 km altitude showed only a 20 % overestimate of nitrate (Fig. 1). This is because the model bias is mainly driven by nighttime conditions (Fig. 5), while aircraft samples in the daytime during KORUS-AQ. The cause of this large model bias is analyzed by Travis et al. (2021) and is attributed to nighttime nitrate chemistry and deposition in the stratified boundary layer.

## 5 AOD and its relationship to $PM_{2.5}$ over East Asia

We build on our analysis of the KORUS-AQ period for a broader interpretation of the distribution of AOD over Korea and China and its relationship to surface $PM_{2.5}$, acknowledging that the conditions sampled in KORUS-AQ may not be representative of other seasons or of China. Figure 6 shows the spatial distributions of 2016 annual and seasonal mean geostationary (GEO) satellite AODs, the corresponding GEOS-Chem clear-sky AODs, and GEOS-Chem surface $PM_{2.5}$. The figure gives normalized mean biases (NMBs) relative to ground-based measurements from AERONET and from the $PM_{2.5}$ surface networks (shown as circles) over the North China region (34.5–40.5° N, 115.5–122° E) and South Korea. The North China region is defined to overlap with the

domain of the geostationary satellite AOD and to ensure consistent seasonal variations within its narrow latitude.

On an annual mean basis, AOD over North China ($\sim 0.5$–0.6) is about 50 % larger than over South Korea ($\sim 0.3$–0.4). AOD over South Korea shows higher values ($> 0.4$) in the Seoul metropolitan area, consistent with that during the KORUS-AQ period (Fig. 4a). Transport from the Asian continent is strongest in spring, when the frequency of cold front passages is highest (Liu et al., 2003). AERONET total AOD in spring (0.4–0.6) is twice as large as fine-mode AOD (0.2–0.3), reflecting a large contribution of dust. In seasons other than spring, 80 %–90 % of AERONET total AOD is contributed by the fine mode. There is large seasonality in AODs over North China and weaker seasonality over South Korea, which is discussed below.

The GEOS-Chem clear-sky AODs show the same spatial and seasonal patterns as GEO satellite AODs but tend to be low in spring and summer. Comparison of the model to AERONET AODs confirms this bias and shows better agreement with fine-mode AOD in spring (NMB of −1 %), implying an underestimate of coarse dust that is consistent with our comparisons to the AirKorea network data during KORUS-AQ (Fig. 4e). Comparison of clear-sky and all-sky AODs in GEOS-Chem shows no significant difference on an annual and seasonal mean basis, except for winter (Fig. S7). Winter has larger all-sky AOD than clear-sky AOD and the lowest rate of successful satellite retrievals (Fig. S7), which may be due in part to misclassification of heavy wintertime $PM_{2.5}$ pollution as clouds (Zhang et al., 2020).

The spatial distributions of $PM_{2.5}$ in GEOS-Chem in different seasons match closely the observations (Fig. 6, bottom row). We see also a close coincidence between the spatial distributions of $PM_{2.5}$ and AODs, in both the observations and the model. On an annual mean basis, GEOS-Chem overesti-

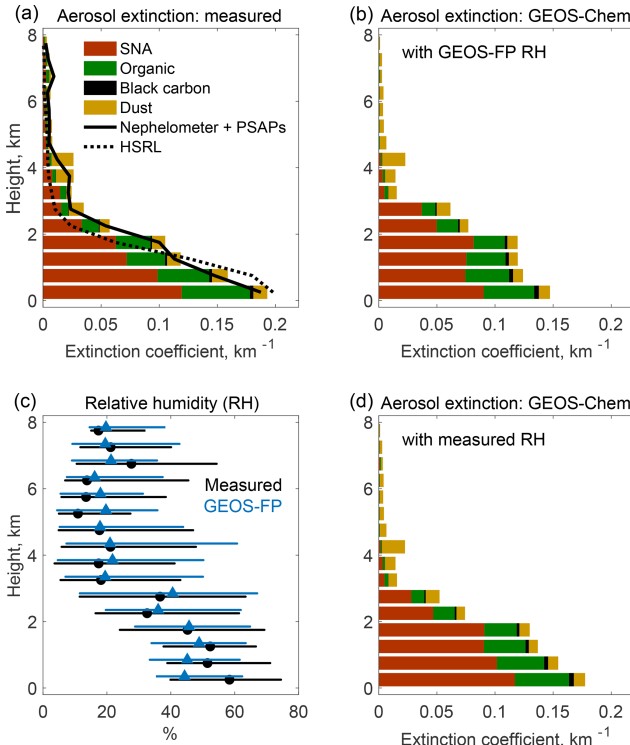

**Figure 3.** Vertical profiles of aerosol extinction coefficients and relative humidity (RH) during KORUS-AQ. Panel **(a)** shows the mean observed vertical distributions of 550 nm extinction coefficients measured in situ (nephelometer + PSAPs; at ambient RH) and remotely (HSRL), along with an independent calculation (colored horizontal bars) from the measured mass concentrations of major aerosol components, measured RH, and GEOS-Chem optical properties as given in Table 3. Panel **(b)** shows the mean aerosol extinction profile in GEOS-Chem and the contributions from the different model components. Panel **(c)** is the median vertical profile of RH (horizontal bars are 25th–75th percentiles) from aircraft measurements and the GEOS-FP assimilated meteorological data used to drive GEOS-Chem. Panel **(d)** is the same as **(b)** but calculated using measured RH.

mates $PM_{2.5}$ by 16 % in North China and by 14 % in South Korea, even though it underestimates AERONET fine-mode AODs by 15 %. The overestimate of $PM_{2.5}$ in South Korea is worst in spring (27 %), consistent with KORUS-AQ results, which we previously attributed to excessive nighttime nitrate build-up in the model. Over North China, the overestimate of $PM_{2.5}$ is worst in summer (33 %), consistent with the nitrate overestimate in summer shown in our previous study (Zhai et al., 2021), which could also be due to model overestimate of nighttime nitrate (Miao et al., 2020).

Figure 7 shows daily correlations of the regional average series between AERONET total AOD and GEO satellite AOD, between AERONET fine-mode AOD and GEOS-Chem AOD, and between measured $PM_{2.5}$ and GEOS-Chem $PM_{2.5}$. Correlations in Fig. 7 are all statistically significant, with correlation coefficients ($R$) ranging from around 0.7 to

more than 0.9 and normalized mean biases (NMB) within $\pm 30$ %. The correlations of these three pairs are similar over South Korea and North China, except that GEOS-Chem overestimates springtime $PM_{2.5}$ in South Korea but not over North China, possibly due to a model overestimate of the long-range transport of $PM_{2.5}$ from China to South Korea in spring.

Figure 8 compares the seasonalities of AOD and $PM_{2.5}$ over the North China and South Korea regions. The GEO satellite AOD over North China peaks in July and is minimum in winter. Most of the AOD is attributed by GEOS-Chem to SNA aerosol, same as in South Korea. AOD over South Korea also has a summer maximum and winter minimum but with weaker amplitude than over North China. The GEOS-Chem AOD is biased low by $\sim 20$ % in summer, and this is largely due to a low RH bias (Fig. S8), as seen previously in the KORUS-AQ comparisons, but amplified by the high RH in summer that drives hygroscopic growth (Latimer and Martin, 2019).

Surface $PM_{2.5}$ in the observations over North China and South Korea shows opposite seasonality to AOD, with minimum values in summer and maximum values in winter–spring. GEOS-Chem reproduces the strong seasonality of $PM_{2.5}$ in North China and the much weaker seasonality in South Korea. The high $PM_{2.5}$ values over North China in winter in the model are mostly driven by organic aerosol, reflecting the large residential coal burning source (Fig. S9; Zheng et al., 2018). In South Korea, by contrast, household energy is mainly from natural gas and electricity (Lee et al., 2020; Woo et al., 2020). GEOS-FP daytime PBL height also shows a stronger seasonality over North China than over South Korea (Fig. S8), generally consistent with the CALIPSO daytime PBL height (Su et al., 2018). Previous studies have shown opposite seasonality between MODIS AOD and surface $PM_{2.5}$ over North China and attributed this to the seasonality in PBL height and RH (Qu et al., 2016; Xu et al., 2019). The mean $PM_{2.5}$ / AOD ratio over North China in winter (236 µg m$^{-3}$) is 8 times that in summer (29 µg m$^{-3}$), with autumn (94 µg m$^{-3}$) and spring (89 µg m$^{-3}$) in between, while over South Korea, the $PM_{2.5}$ / AOD ratio in winter (62 µg m$^{-3}$) is only 70 % larger than in summer (36 µg m$^{-3}$).

## 6   Conclusions

Geostationary satellite observations of aerosol optical depth (AOD) over East Asia may usefully complement $PM_{2.5}$ air quality networks if the local relationship between AOD and $PM_{2.5}$ can be inferred from a physical and/or statistical model. Here we analyzed the ability of the GEOS-Chem chemical transport model to provide this relationship by using a new fused GOCI–AHI geostationary satellite product together with AERONET ground-based AOD measurements, aerosol vertical profiles over South Korea from the KORUS-

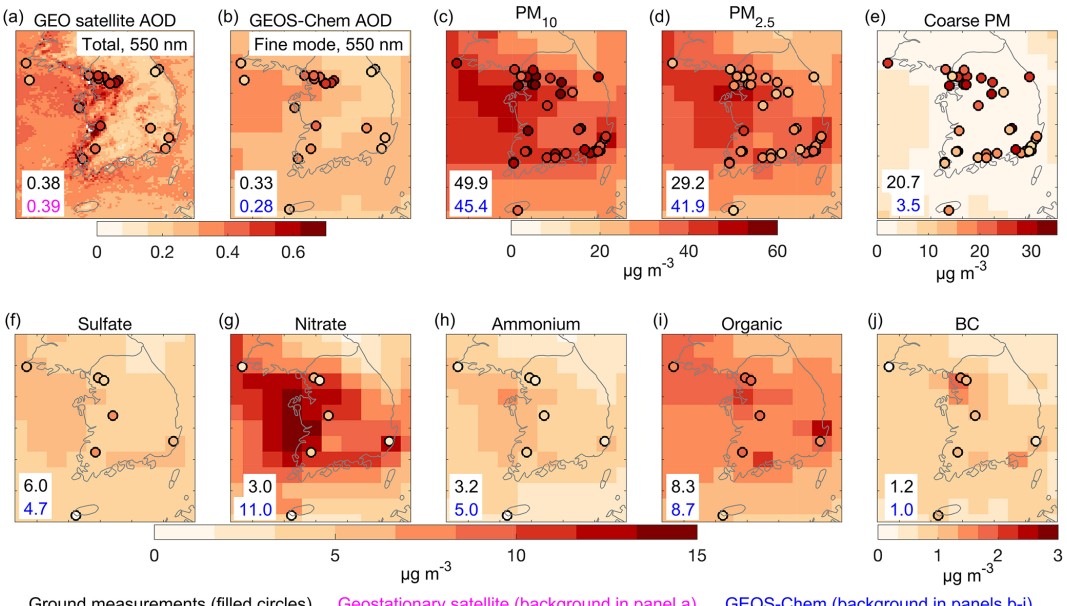

**Figure 4.** Spatial distributions of AOD and surface $PM_{10}$, $PM_{2.5}$, coarse PM ($PM_{10}$ minus $PM_{2.5}$), and major $PM_{2.5}$ components over South Korea averaged during KORUS-AQ (9 May–10 June 2016). Panel **(a)** shows the fused geostationary (GEO) 550 nm AOD from the GOCI and AHI satellites (background) and AERONET 550 nm total AOD (filled circles). Panel **(b)** shows GEOS-Chem 550 nm AOD sampled at hourly GEO satellite AOD (GEOS-Chem clear-sky AOD; background) and AERONET 550 nm fine-mode AOD (filled circles). Panel **(c)** shows surface $PM_{10}$ modeled by GEOS-Chem (background) and measured at ground sites (filled circles). Panels **(d)**–**(j)** are the same as panel **(c)** but respectively for $PM_{2.5}$; coarse PM ($PM_{10}$ minus $PM_{2.5}$); and sulfate, nitrate, ammonium, organic aerosol, and BC $PM_{2.5}$ components. Values inset are median values from ground-based measurements (black) and sampled from GEO satellite (magenta) and GEOS-Chem (blue). Measured $PM_{10}$, $PM_{2.5}$, and coarse PM in panels **(c)**–**(e)** are shown for a random selection of 50 % of AirKorea sites to visualize spatial distribution, and inset values are for the seven supersites where $PM_{2.5}$ composition was measured. Median AOD values inset are sampled at or near the seven supersites to avoid biasing by the large number of sites in the Seoul metropolitan area. Modeled total $PM_{2.5}$ concentrations are calculated at 35 % RH (Table 3). Modeled $PM_{10}$ is the sum of $PM_{2.5}$, coarse dust, and coarse sea salt.

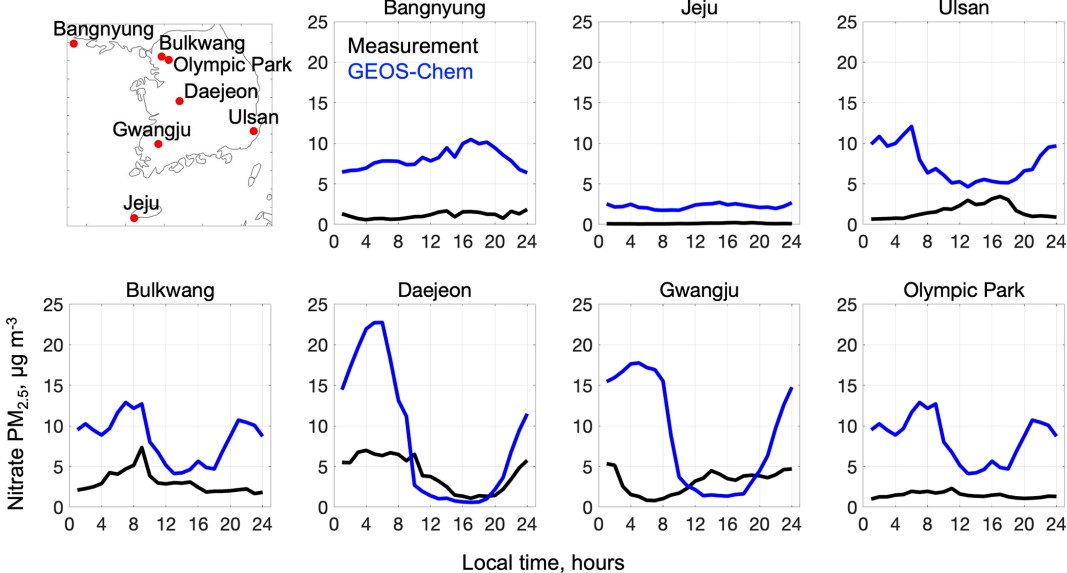

**Figure 5.** Median diurnal variations in $PM_{2.5}$ nitrate concentrations at the seven supersites (top left panel) operated in South Korea during KORUS-AQ (9 May–10 June 2016). Values are medians binned by hour. GEOS-Chem model values are sampled to coincide with the measurements.

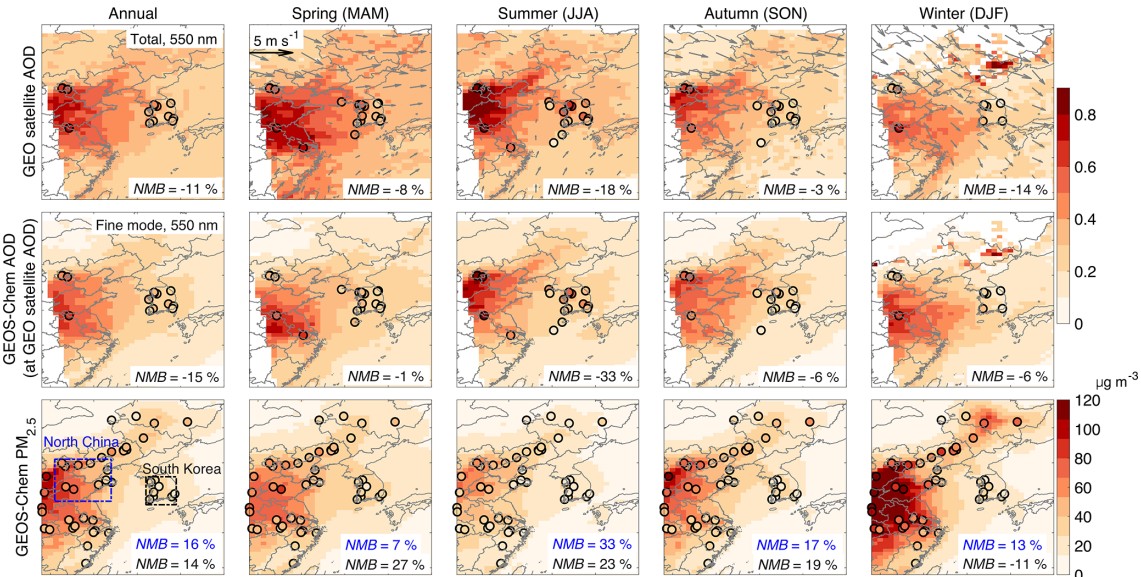

**Figure 6.** Spatial distributions of 2016 annual and seasonal mean AOD (550 nm) and surface PM$_{2.5}$. The top row shows the observed GOCI–AHI geostationary satellite AOD (GEO satellite AOD) on the GEOS-Chem 0.5° × 0.625° grid with superimposed 925 hPa GEOS-FP wind fields and AERONET total AODs (circles). The middle row shows clear-sky GEOS-Chem AOD, with AERONET fine-mode AOD added as circles. The bottom row shows GEOS-Chem surface PM$_{2.5}$ (background) with surface network measurements (circles). AERONET AODs are shown only when more than 10 months of data are available for the annual mean, and all 3 months of data are available for each season. The PM$_{2.5}$ observations shown are for a random selection of 7 % of network sites for visual clarity. GEOS-Chem PM$_{2.5}$ is calculated at 35 % RH (Table 3). Normalized mean biases (NMBs) inset are for the comparisons of GEO satellite and GEOS-Chem values to the corresponding ground measurements.

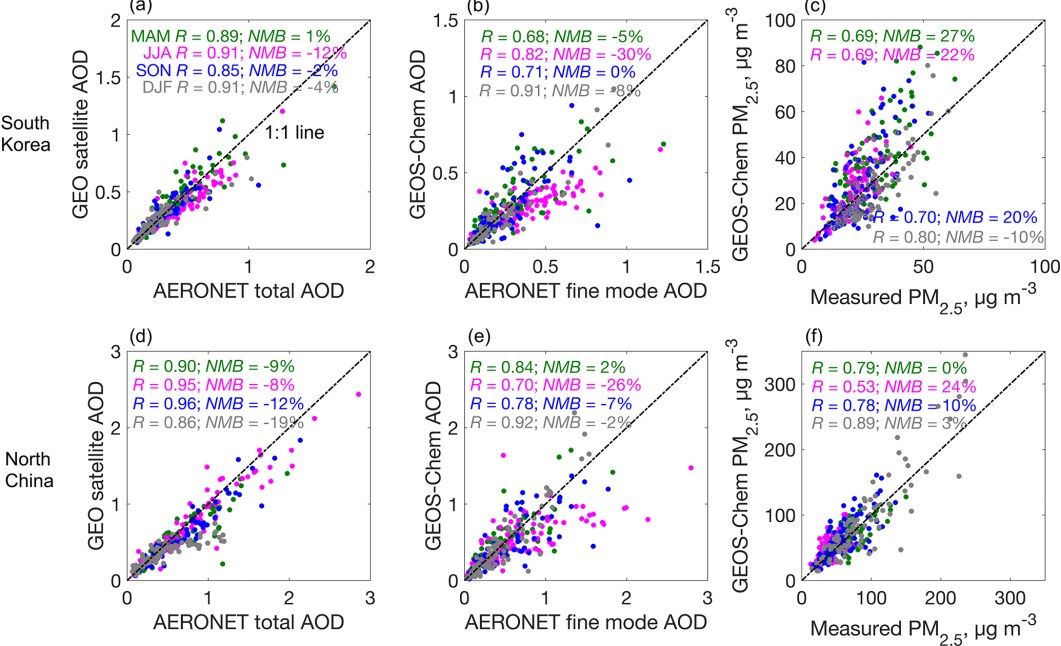

**Figure 7.** Scatterplots of regional mean daily **(a, d)** GEO satellite AOD vs. AERONET total AOD, **(b, e)** GEOS-Chem AOD vs. AERONET fine-model AOD, and **(c, f)** GEOS-Chem PM$_{2.5}$ vs. measured PM$_{2.5}$ over South Korea **(a–c)** and North China **(d–f)**. Different colors represent different seasons. Values inset are correlation coefficients ($R$) and normalized mean biases (NMBs) between surface measurements and GEO satellite or GEOS-Chem values.

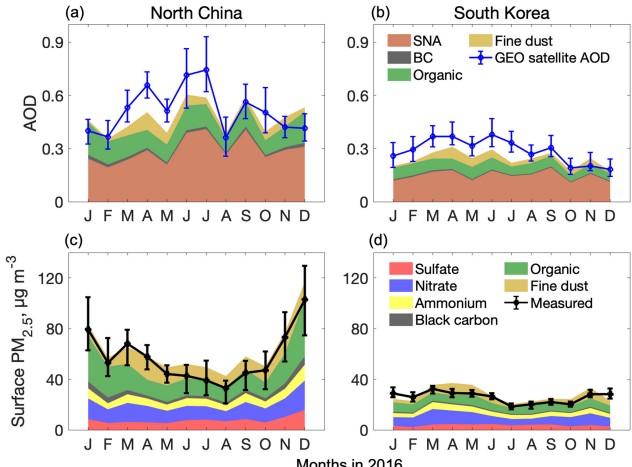

**Figure 8.** Seasonality of AOD and PM$_{2.5}$ over North China and South Korea and contributions from individual aerosol components. Lines show regional medians (error bars: 25th and 75th percentiles) for the ensemble of monthly averaged observations in the regions (Fig. 6) in 2016. GEOS-Chem values are shown as stacked contours for individual components and are sampled in the same way as the observations.

AQ aircraft campaign (May–June 2016), and surface network observations. This allowed us to identify the critical features and limitations of the model for successfully representing the AOD–PM$_{2.5}$ relationship.

The KORUS-AQ observations show that total aerosol extinction (550 nm) in the vertical column is dominated by sulfate–nitrate–ammonium (SNA) and organic aerosol in the planetary boundary layer (PBL), despite large concentrations of dust in the free troposphere. This reflects the optically effective size and high hygroscopicity of the PBL aerosols. We find that GEOS-Chem aerosol optical properties based on measurements over the North America (default model setting) underestimate KORUS-AQ aerosol mass extinction efficiency by around 20 %. In addition, a low bias in GEOS-FP RH below 1 km leads to a 10 % underestimate of AOD inferred from the aircraft profile. Adjustments of GEOS-Chem aerosol optical properties and RH enable a successful simulation of the aerosol extinction profile. SNA aerosol contributes 59 % of column aerosol extinction in the KORUS-AQ data, while organic aerosol contributes 27 %, and dust contributes 12 %.

Comparison of GOCI–AHI geostationary (GEO) satellite AOD to AERONET AODs over South Korea shows good agreement, with high values along the west coast. GEOS-Chem is more consistent with the fine-mode AERONET AOD because of its insufficient accounting of coarse particles, which account for 13 % of AERONET AOD. The remaining 15 % underestimate of AERONET fine-mode AOD by GEOS-Chem can be attributed to the RH low bias. GEOS-Chem overestimates 24 h surface PM$_{2.5}$ over South Korea by 43 % during the KORUS-AQ period, despite its success-

ful simulation of the aircraft data and fine-mode AERONET AOD, and we find that this is due to a large overestimate of nighttime nitrate.

Broader examination of the GOCI–AHI AOD satellite data over East Asia shows spatial distributions and magnitudes consistent with AERONET and featuring in particular strong Asian outflow in spring that includes a large dust component. We find that AODs and PM$_{2.5}$ have similar large-scale spatial distributions but opposite seasonality. PM$_{2.5}$ in North China has a strong winter maximum and summer minimum, while AOD shows the opposite. GEOS-Chem simulates successfully the seasonality of measured PM$_{2.5}$ but is biased low by $\sim 20$ % in summer for AOD, due again to low RH bias like that during KORUS-AQ, amplified by the high RH in summer that drives hygroscopic growth (Latimer and Martin, 2019). We find that the opposite AOD and PM$_{2.5}$ seasonality is mainly driven by residential coal heating sources and low PBL depths in winter and high RH in summer. Observations of PM$_{2.5}$ and AOD in South Korea show the same seasonal phases as in North China but with much weaker amplitude, reflecting the lack of residential coal burning in winter and a weaker seasonal amplitude of PBL depth.

In summary, we find that the geostationary GOCI–AHI satellite AOD data provide high-quality information for monitoring of PM$_{2.5}$ over East Asia but that physical interpretation requires accurate information on aerosol size distributions, PBL depths, RH, the role of coarse particles, and diurnal variation in PM$_{2.5}$, all of which are subject to large uncertainties in chemical transport models. Addressing these uncertainties should be a target of future work. We have used results from our study in a recent machine-learning reconstruction of daily 2011–present PM$_{2.5}$ over East Asia from GOCI AOD data by identifying critical variables for the machine-learning algorithm and providing blended gap-filling data for cloudy scenes (Pendergrass et al., 2021). Besides the factors discussed in this study, topography might be another important factor influencing surface PM$_{2.5}$ and its vertical mixing (Su et al., 2018), and this also requires future investigation.

*Data availability.* Aircraft data during KORUS-AQ are available at https://doi.org/10.5067/Suborbital/KORUSAQ/DATA01 (Aknan and Chen, 2019). PM$_{2.5}$ data over China are from http://www.quotsoft.net/air/ (CNEMC, 2021). PM$_{2.5}$ data over South Korea are from https://www.airkorea.or.kr/web/last_amb_hour_data?pMENU_NO=123 (KEC, 2021). AERONET data can be found at https://aeronet.gsfc.nasa.gov/ (Giles and Holben, 2014). The MEIC emission inventory is at http://www.meicmodel.org/ (Tsinghua University, 2021). The KORUSv5 emission inventory is developed by Konkuk University, available at http://aisl.konkuk.ac.kr/#/emission_data/korus-aq_emissions (CAIS, 2021).

*Supplement.* The supplement related to this article is available online at: https://doi.org/10.5194/acp-21-1-2021-supplement.

*Author contributions.* SZ and DJJ designed the study. SZ performed the data analysis and model simulations with contributions from JFB, KL, HCL, SKK, XW, PL, KRT, and HoL. JK, SL, and HyL provided satellite AOD data. RJP and JIJ contributed to AirKorea data processing. JM and RM provided the dust emission inventory. GL, FY, and JMM updated wet deposition simulation. JWH, BEA, JED, JLJ, PCJ, and BAN contributed to KORUS-AQ campaign measurements. JHW and YK provided the KORUSv5 emission inventory. QZ provided the MEIC emission inventory. SZ and DJJ wrote the paper with input from all authors.

*Competing interests.* The contact author has declared that neither they nor their co-authors have any competing interests.

*Financial support.* This work was funded by the Samsung Advanced Institute of Technology and the Harvard–NUIST Joint Laboratory for Air Quality and Climate (JLAQC). Jose L. Jimenez, Pedro Campuzano-Jost, and Benjamin A. Nault were supported by NASA (grant nos. NNX15AT96G and 80NSSC19K0124).

*Review statement.* This paper was edited by Zhanqing Li and reviewed by three anonymous referees.

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
