# Peer review of "Relating geostationary satellite measurements of aerosol optical"

_Atmospheric Chemistry and Physics, 2021_

## Author Response (AR1)

We acknowledge the referees for their insightful comments. We have made efforts to improve the manuscript accordingly. Please find our responses to referees' comments in blue. Newly added references are listed at the end of this document.

**RC1 by Referee #3**

This paper attempts to understand the relationship between AOD and $PM_{2.5}$. However, after reading through, I feel that the paper is more of a GEOS-Chem validation and uncertainty analysis work, rather than offering physical explanation of the AOD-$PM_{2.5}$ relationship.

The title has been revised to: "Relating geostationary satellite measurements of aerosol optical depth (AOD) over East Asia to fine particulate matter ($PM_{2.5}$): insights from the KORUS-AQ aircraft campaign and GEOS-Chem model simulations".

We have rephrased lines 34-38 in the abstract: "Geostationary satellite measurements of aerosol optical depth (AOD) over East Asia from the GOCI and AHI instruments can augment surface monitoring of fine particulate matter ($PM_{2.5}$) air quality, but this requires better understanding of the AOD-$PM_{2.5}$ relationship. Here we use the GEOS-Chem chemical transport model to analyze the critical variables determining the AOD-$PM_{2.5}$ relationship over East Asia by simulation of observations from satellite, aircraft, and ground-based datasets."

We added lines 67-68: "This enables us to identify critical variables and uncertainties for inferring $PM_{2.5}$ from satellite AOD data."

We rephrased lines 433-439 in the conclusions section: "Geostationary satellite observations of aerosol optical depth (AOD) over East Asia may usefully complement $PM_{2.5}$ air quality networks if the local relationship between AOD and $PM_{2.5}$ can be inferred from a physical and/or statistical model. Here we analyzed the ability of the GEOS-Chem chemical transport model to provide this relationship by using a new fused GOCI/AHI geostationary satellite product together with AERONET ground-based AOD measurements, aerosol vertical profiles over South Korea from the KORUS-AQ aircraft campaign (May-June 2016), and surface network observations. This allowed us to identify the critical features and limitations of the model for successful representing the AOD-$PM_{2.5}$ relationship."

Specifically, could the authors clarify, perhaps with additional analysis, how different factors, such as PBL height, RH, organic matter fraction, etc, contribute to the uncertainty in AOD-PM$_{2.5}$ relationship? How does the role of each factor vary with region (e.g., Korea vs. China)? The only clear conclusion is that AOD and PM$_{2.5}$ have reversed seasonality because of seasonally varying PBL height, but this is already well known.

We quantified in the abstract (lines 43-47): "We updated SNA and organic aerosol size distributions in GEOS-Chem to represent aerosol optical properties over East Asia by using in-situ measurements of particle size distributions from KORUS-AQ. We find that SNA and organic aerosols over East Asia have larger size (number median radius of 0.11 μm with geometric standard deviation of 1.4) and 20% larger mass extinction efficiency as compared to aerosols over North America (default setting in GEOS-Chem)."

We quantified in lines 286-287: "The model underestimates extinction coefficients by 20% below 1 km altitude, leading to a 10% underestimate of aircraft inferred AOD, although there is no such underestimate in aerosol mass."

We quantified in lines 323-324 to: "Therefore, about half of the GEOS-Chem underestimate of total AOD can be attributed to missing coarse PM, with the other half comes from negative RH bias."

We added a Figure 7 and lines 399-402: "The correlations of these three pairs are similar over South Korea and North China, except that GEOS-Chem overestimates springtime PM$_{2.5}$ in South Korea but not over North China, possibly due to a model overestimate of the long-range transport of PM$_{2.5}$ from China to South Korea in spring."

[Figure]

**Figure 7. Scatter plots of regional mean daily (a and d) GEO satellite AOD vs. AERONET total AOD, (b and e) GEOS-Chem AOD vs. AERONET fine-model AOD, and (c and f) GEOS-Chem PM2.5 vs. measured PM2.5 over South Korea (a-c) and North China (d-f). Different colors represent different seasons. Values inset are correlation coefficients (R) and normalized mean biases (NMB) between surface measurements and GEO satellite or GEOS-Chem values.**

We quantified in lines 412-414: "The GEOS-Chem AOD is ~ 20% biased low in summer and this is largely due to a low RH bias (Figure S8), as seen previously in the KORUS-AQ comparisons but amplified by the high RH in summer that drives hygroscopic growth (Latimer and Martin, 2019)."

We added analysis in lines 420-423: "GEOS-FP daytime PBL height also shows a stronger seasonality over North China than over South Korea (Figure S8), generally consistent with the CALIPSO daytime PBL height (Su et al., 2018). Previous studies have shown opposite seasonality between MODIS AOD and surface PM$_{2.5}$ over North China and attributed this to the seasonality in PBL height and RH (Qu et al., 2016; Xu et al., 2019)."

We quantified in lines 443-446 in the conclusion section: "We find that GEOS-Chem aerosol optical properties based on measurements over the North America (default model setting) underestimate KORUS-AQ aerosol mass extinction efficiency by around 20%. In addition, a low bias in GEOS-FP RH below 1 km leads to a 10% underestimate of AOD inferred from the aircraft profile."

We added lines 451-452 in the conclusions section: "The remaining 15% underestimate of AERONET fine-mode AOD by GEOS-Chem can be attributed to the RH low bias."

We quantified in lines 460-462 in the conclusions section: "GEOS-Chem simulates successfully the seasonality of measured PM2.5 but is ~ 20% biased low in summer for AOD, due again to RH low bias like that during KORUS-AQ, amplified by the high RH in summer that drives hygroscopic growth (Latimer and Martin, 2019)."

**RC2 by Referee #2**

The manuscript investigated the physical relationships between AOD and PM$_{2.5}$ over East Asia by using the model simulation and comprehensive observation. The results indicate that the aerosols over this region are largely contributed by the sulfate-nitrate-ammonium and organic aerosols within the PBL. Meanwhile, the dust in the free troposphere also has an important contribution to column AOD. The seasonality of AOD and PM$_{2.5}$ has been specifically discussed. In general, this paper is well-written with a good logical connection. Thus, I recommend the manuscript for publication in Atmospheric Chemistry and Physics, after addressing the following comments.

Specific Comments:

1.      The current introduction section may be insufficient to demonstrate the significance of this paper. The authors need to clearly explain the limitation of previous studies and the advantage of this study.

We revised the whole introduction:

"$PM_{2.5}$ (particulate matter with aerodynamic diameter less than 2.5 µm) in surface air is a severe public health concern in East Asia, but surface monitoring networks are too sparse to thoroughly assess population exposure. Satellite observations of aerosol optical depth (AOD) can provide a valuable complement (Van Donkelaar et al., 2015). Geostationary satellite sensors, including the Geostationary Ocean Color Imager (GOCI) launched by the Korea Aerospace Research Institute (KARI) in 2011 (Choi et al., 2016, 2018, 2019) and the Advanced Himawari Imager (AHI) launched by the Japanese Meteorological Agency (JMA) in 2014 (Lim et al., 2018, 2021), offer the potential for high-density mapping of $PM_{2.5}$ over East Asia. However, more confidence is needed in relating AOD to $PM_{2.5}$. Here we evaluate the capability of the GEOS-Chem chemical transport model (CTM) to simulate AOD-$PM_{2.5}$ relationships over East Asia, exploiting in-situ aircraft measurements of vertical aerosol profiles and optical properties from the joint NASA-NIER Korea - United States Air Quality (KORUS-AQ) field study in May-June 2016 (Crawford et al., 2021; Peterson et al., 2019; Jordan et al., 2020) together with GOCI/AHI geostationary satellite data and surface measurement networks. This enables us to identify critical variables and uncertainties for inferring $PM_{2.5}$ from satellite AOD data.

A number of past studies have used satellite AOD data to infer surface $PM_{2.5}$ using physical and statistical models. The standard geophysical approach has been to use a CTM, such as GEOS-Chem, to compute the $PM_{2.5}$/AOD ratio (Liu et al., 2004; van Donkelaar et al., 2006; van Donkelaar et al., 2015; Xu et al., 2015; Geng et al., 2017), with recent applications correcting for CTM biases using available $PM_{2.5}$ surface network data (Brauer et al., 2016; Van Donkelaar et al., 2016; van Donkelaar et al., 2019; Hammer et al., 2020). An alternative approach is to use machine-learning algorithms to relate satellite AOD to $PM_{2.5}$ by training on the surface network data (Hu et al., 2017; Chen et al., 2018; Xiao et al., 2018; Wei et al., 2021; Pendergrass et al., 2021), and sometimes including CTM values as predictors (Di et al., 2019; Xue et al., 2019). Yet another approach is to assimilate the satellite-measured AODs in a CTM and correct in this manner the $PM_{2.5}$ simulation, although this requires attribution of model AOD errors to specific model parameters (Kumar et al., 2019; Saide et al., 2014; Sekiyama et al., 2010; Cheng et al., 2019). In all of these approaches, a better physical

understanding of the AOD-PM$_{2.5}$ relationship as simulated by CTMs can greatly enhance the capability to infer PM¬2.5 from AOD data.

AOD measures aerosol extinction (scattering and absorption) integrated over the atmospheric column, so that its relationship to 24-hr average surface PM$_{2.5}$ (the standard air quality metric) depends on the aerosol vertical distribution and optical properties, ambient relative humidity (RH), diurnal variation of PM$_{2.5}$, and contribution from coarse particulate matter to AOD. Little study of these factors has been conducted for East Asia. Airborne measurements of aerosol vertical profiles in East Asia are very limited (Liu et al., 2009; Sun et al., 2013). AOD is highly sensitive to RH (Brock et al., 2016; Latimer and Martin et al., 2019; Saide et al., 2020), but the impact from RH uncertainty on AOD simulation lacks evaluation. In addition, because the AOD is a daytime measurement that needs to be related to 24-h average PM$_{2.5}$, the diurnal variation of PM$_{2.5}$ needs to be understood (Guo et al., 2017; Lennartson et al., 2018). Finally, there has been to our knowledge no study of how coarse anthropogenic PM may contribute to the AOD measurements. Coarse anthropogenic PM (distinct from desert dust) is known to be high over East Asia (Chen et al., 2015; Dai et al., 2018)."

We revised lines 191-195: "Therefore, we re-computed the diagnostic AOD using updated log-normal size distributions for SNA and organic aerosol with number median radius $R_{N,med}$ = 0.11 µm and geometric standard deviation $\sigma$ = 1.4 based on KORUS-AQ observations, instead as compared tof $R_{N,med}$ = 0.058 µm and $\sigma$ = 1.6 in the standard model version 12.7.1, which is derived from IMPROVE network measurements of aerosol mass scattering efficiency over North America (Latimer and Martin, 2019)."

2.      The analyses of this study are closely associated with the model simulation of GEOS-Chem, while the title only mentioned the observations. There are some disconnections between the title and the main text.

The title has been revised to: "Relating geostationary satellite measurements of aerosol optical depth (AOD) over East Asia to fine particulate matter (PM$_{2.5}$): insights from the KORUS-AQ aircraft campaign and GEOS-Chem model simulations".

3.      Line 203, Page 8. The PBL varies significantly during the different periods. It is risky to define the 0-2 km as the PBL. The authors should give more justifications for this definition.

We rewrote line 221 to: "…, which we define as the average planetary boundary layer (PBL) during KORUS-AQ, …"

We have line 232: "KORUS-AQ aerosol component profiles for different meteorological regimes is presented in Park et al. (2021)."

4.      The seasonality of AOD and PM$_{2.5}$ and its association with PBLH have been discussed previously (e.g., Guo et al., 2017; Su et al., 2018). I suggest the authors acknowledge these works.

Introduction

It is too short and the authors are suggested to summarize previous studies on investigating the relationships between $PM_{2.5}$ and AOD, especially those focusing on Asia.

In addition, studies on PM estimation from satellite AOD products need to be summarized, especially those using geostationary satellites.

Finally, the author should highlight the innovation and difference between the current study and previous related studies, and discuss the importance of understanding the relationships between PM and AOD in these studies.

The revised introduction is pasted as below:

"$PM_{2.5}$ (particulate matter with aerodynamic diameter less than 2.5 μm) in surface air is a severe public health concern in East Asia, but surface monitoring networks are too sparse to thoroughly assess population exposure. Satellite observations of aerosol optical depth (AOD) can provide a valuable complement (Van Donkelaar et al., 2015). Geostationary satellite sensors, including the Geostationary Ocean Color Imager (GOCI) launched by the Korea Aerospace Research Institute (KARI) in 2011 (Choi et al., 2016, 2018, 2019) and the Advanced Himawari Imager (AHI) launched by the Japanese Meteorological Agency (JMA) in 2014 (Lim et al., 2018, 2021), offer the potential for high-density mapping of $PM_{2.5}$ over East Asia. However, more confidence is needed in relating AOD to $PM_{2.5}$. Here we evaluate the capability of the GEOS-Chem chemical transport model (CTM) to simulate AOD-$PM_{2.5}$ relationships over East Asia, exploiting in-situ aircraft measurements of vertical

aerosol profiles and optical properties from the joint NASA-NIER Korea - United States Air Quality (KORUS-AQ) field study in May-June 2016 (Crawford et al., 2021; Peterson et al., 2019; Jordan et al., 2020) together with GOCI/AHI geostationary satellite data and surface measurement networks. This enables us to identify critical variables and uncertainties for inferring $PM_{2.5}$ from satellite AOD data.

A number of past studies have used satellite AOD data to infer surface $PM_{2.5}$ using physical and statistical models. The standard geophysical approach has been to use a CTM, such as GEOS-Chem, to compute the $PM_{2.5}$/AOD ratio (Liu et al., 2004; van Donkelaar et al., 2006; van Donkelaar et al., 2015; Xu et al., 2015; Geng et al., 2017), with recent applications correcting for CTM biases using available $PM_{2.5}$ surface network data (Brauer et al., 2016; Van Donkelaar et al., 2016; van Donkelaar et al., 2019; Hammer et al., 2020). An alternative approach is to use machine-learning algorithms to relate satellite AOD to $PM_{2.5}$ by training on the surface network data (Hu et al., 2017; Chen et al., 2018; Xiao et al., 2018; Wei et al., 2021; Pendergrass et al., 2021), and sometimes including CTM values as predictors (Di et al., 2019; Xue et al., 2019). Yet another approach is to assimilate the satellite-measured AODs in a CTM and correct in this manner the $PM_{2.5}$ simulation, although this requires attribution of model AOD errors to specific model parameters (Kumar et al., 2019; Saide et al., 2014; Sekiyama et al., 2010; Cheng et al., 2019). In all of these approaches, a better physical understanding of the AOD-$PM_{2.5}$ relationship as simulated by CTMs can greatly enhance the capability to infer PM¬2.5 from AOD data.

AOD measures aerosol extinction (scattering and absorption) integrated over the atmospheric column, so that its relationship to 24-hr average surface $PM_{2.5}$ (the standard air quality metric) depends on the aerosol vertical distribution and optical properties, ambient relative humidity (RH), diurnal variation of $PM_{2.5}$, and contribution from coarse particulate matter to AOD. Little study of these factors has been conducted for East Asia. Airborne measurements of aerosol vertical profiles in East Asia are very limited (Liu et al., 2009; Sun et al., 2013). AOD is highly sensitive to RH (Brock et al., 2016; Latimer and Martin et al., 2019; Saide et al., 2020), but the impact from RH uncertainty on AOD simulation lacks evaluation. In addition, because the AOD is a daytime measurement that needs to be related to 24-h average $PM_{2.5}$, the diurnal variation of $PM_{2.5}$ needs to be understood (Guo et al., 2017; Lennartson et al., 2018). Finally, there has been to our knowledge no study of how coarse anthropogenic PM may contribute to the AOD measurements. Coarse anthropogenic PM (distinct from desert dust) is known to be high over East Asia (Chen et al., 2015; Dai et al., 2018)."

We revised lines 191-195: "Therefore, we re-computed the diagnostic AOD using updated log-normal size distributions for SNA and organic aerosol with number median radius $R_{N,med}$ = 0.11 μm and

geometric standard deviation $\sigma$ = 1.4 based on KORUS-AQ observations, instead as compared tof $R_{N,med}$ = 0.058 μm and $\sigma$ = 1.6 in the standard model version 12.7.1, which is derived from IMPROVE network measurements of aerosol mass scattering efficiency over North America (Latimer and Martin, 2019)."

Lines 85-86: Ångström Exponents at 500 nm? AE refers to a wavelength range. Reference is needed here.

We detailed in lines 98-101: "We use total and fine-mode AODs at 500 nm wavelength from the AERONET Version 3; Spectral Deconvolution Algorithm (SDA) Version 4.1 Retrieval Level 2.0 database (Giles et al., 2019; O'Neill et al., 2003). The AERONET AODs at 500 nm are converted to 550 nm using total and fine mode Ångström Exponents at 500 nm for consistency with the satellite AOD data."

Lines 107-110: Himawari-8/AHI: Which version do you use? Reference is needed. Again, Himawari-8/AHI provides AOD products at 500 nm. I am not sure about GOCI (should be 550 nm). Are they the same? If not, does the wavelength difference be taken into account in the data fusion?

We detailed in lines 122-126: "Geostationary satellite AOD at 550 nm are retrieved by the Yonsei Aerosol Retrieval (YAER) algorithm for the GOCI (Choi et al., 2016, 2018) and AHI (Lim et al. 2018) instruments, with GOCI covering East China and South Korea and AHI covering the broad East Asia region. AOD from GOCI and AHI have a 6 km × 6 km spatial resolution and 1-hour (GOCI) to 2.5-minute (AHI) temporal resolution for 8 hours per day (09:30 to 16:30 local time)."

Lines 288-290: What are the potential reasons? Is it the aerosol algorithm or the difference caused by sample matching at different wavelengths?

We explained in lines 308-312: "The low biases in the SMA could be due to high-concentration aerosol pixels mis-identified as clouds and/or possible issues with the aerosol type assumption in the aerosol retrieval, while the high biases on the Yellow Sea islands could result from uncertainties in the assumption of ocean surface reflectance, as has been discussed by Choi et al. (2016, 2018) and Lim et al. (2018, 2021)."

Lines 295-296: What are the potential reasons?

We rephrased lines 317-319: "GEOS-Chem reproduces the satellite AOD enhancements along the west coast of South Korea but the values are lower than observed, which we attribute to unaccounted coarse PM and negative RH bias as discussed below."

We quantified in lines 323-325: "Therefore, about half of the GEOS-Chem underestimate of total AOD can be attributed to missing coarse PM, with the other half comes from negative RH bias."

Lines 297-312: Is there any relevant published literature to support the author's explanations of reasons for these differences between GEOS-Chem and satellites observations?

We added in lines 85-91 in the introduction: "AOD is highly sensitive to RH (Brock et al., 2016; Latimer and Martin et al., 2019; Saide et al., 2020), but the impact from RH uncertainty on AOD simulation lacks evaluation. In addition, because the AOD is a daytime measurement that needs to be related to 24-h average $PM_{2.5}$, the diurnal variation of $PM_{2.5}$ needs to be understood (Guo et al., 2017; Lennartson et al., 2018). Finally, there has been to our knowledge no study of how coarse anthropogenic PM may contribute to the AOD measurements. Coarse anthropogenic PM (distinct from desert dust) is known to be high over East Asia (Chen et al., 2015; Dai et al., 2018)."

I also suggest adding some scatter plots to validate and compare the satellite-based and modeled AODs, $PM_{2.5}$, and other parameters if possible, so that readers can see their differences more clearly.

Lines 396-402: "Figure 7 shows daily correlations of the regional average series between AERONET total AOD and GEO satellite AOD, between AERONET fine mode AOD and GEOS-Chem AOD, as well as between measured $PM_{2.5}$ and GEOS-Chem $PM_{2.5}$. Correlations in Figure 7 are all statistically significant with correlation coefficients ($R$) ranging from around 0.7 to more than 0.9 and normalized mean biases ($NMB$) within ± 30%. The correlations of these three pairs are similar over South Korea and North China, except that GEOS-Chem overestimates springtime $PM_{2.5}$ in South Korea but not over North China, possibly due to a model overestimate of the long-range transport of $PM_{2.5}$ from China to South Korea in spring."

[Figure]

**Figure 7. Scatter plots of regional mean daily (a and d) GEO satellite AOD vs. AERONET total AOD, (b and e) GEOS-Chem AOD vs. AERONET fine-model AOD, and (c and f) GEOS-Chem PM$_{2.5}$ vs. measured PM$_{2.5}$ over South Korea (a-c) and North China (d-f). Different colors represent different seasons. Values inset are correlation coefficients (*R*) and normalized mean biases (*NMB*) between surface measurements and GEO satellite or GEOS-Chem values.**

Figure 6: I suggest adding some satellite PM$_{2.5}$ estimated results to see the difference with model simulations since there are many available PM products, especially in China.

We added lines 470-472 in the conclusion section: "We have used results from our study in a recent machine-learning reconstruction of daily 2011-present PM$_{2.5}$ over East Asia from GOCI AOD data by identifying critical variables for the machine-learning algorithm and providing blended gap-filling data for cloudy scenes (Pendergrass et al., 2021)."

Last, the authors should consider the impact of other factors, especially BLH, meteorological conditions, and topography, on surface PM, and to see how much impact can they have on the differences between satellite and model results.

[revised manuscript text omitted]

---

## Author Response (AR2)

We acknowledge the referees for their insightful comments. Please find our responses to referees' comments in blue.

**Report #1 from Referee #2**

The revised manuscript well addressed most of my concerns. I have a few minor comments.

1. Line 62, Page 2. The authors may add some pertinent studies for high-density mapping of $PM_{2.5}$ over East Asia (e.g., Chen et al., 2019).

Added.

2. Line 84, Page 3. The statement of "little study" may be revised. In fact, there have been many studies in East Asia, as reviewed in Li et al., 2017.

We deleted the sentence. We added reference "Li et al., 2017" and two publications cited by "Li et al., 2017", and revised lines 84-86 to: "Airborne measurements of aerosol vertical profiles (without species information) in East Asia are limited (Zhang et al., 2006; Liu et al., 2009; Zhang et al., 2009; Sun et al., 2013; Li et al., 2017), and speciated vertical profiles are rarer."

References:

Zhang, Q., Zhao, C., Tie, X., Wei, Q., Huang, M., Li, G., Ying, Z., and Li, C.: Characterizations of aerosols over the Beijing region: A case study of aircraft measurements, Atmos. Environ., 40, 4513-4527, https://doi.org/10.1016/j.atmosenv.2006.04.032, 2006.

Zhang, Q., Ma, X., Tie, X., Huang, M., and Zhao, C.: Vertical distributions of aerosols under different weather conditions: Analysis of in-situ aircraft measurements in Beijing, China, Atmos. Environ., 43, 5526-5535, https://doi.org/10.1016/j.atmosenv.2009.05.037, 2009.

3. Line 89, Page 3. The authors may revise the statement. First, this study mainly focused on the $PM_{2.5}$ rather than coarse PM. Second, multiple studies have investigated the contribution of coarse aerosol to AOD, which also can be represented by fine mode fraction (Eck et al., 2010).

We clarified lines 89-91 to: "Finally, although there are studies on the optical depth of coarse mode desert dust (Eck et al., 2010; Ridley et al., 2016), there has been to our knowledge no study of how coarse anthropogenic PM may contribute to the AOD measurements."

Reference:

Ridley, D. A., Heald, C. L., Kok, J. F., and Zhao, C.: An observationally constrained estimate of global dust aerosol optical depth, Atmos. Chem. Phys., 16, 15097-15117, 10.5194/acp-16-15097-2016, 2016.

4. Section 3.1. The authors define 0-2 km as the average PBL. Note that the average daily maximum PBL depth is around 2 km. PBL during the morning and evening is much lower. Does Figure 1 only present the noontime results or the whole day average?

We have added 'aircraft' in line 219 and added a line 220: "The KORUS-AQ aircraft sampled during the daytime, mainly between 9 am and 3 pm local time."

Line 335-337 were revised to: "This is because the model bias is mainly driven by nighttime conditions (Figure 5), while aircraft samples in the daytime during KORUS-AQ."

5. Table 1. The authors may briefly explain why choose these sites, since there are many other PM stations over East Asia.

We have changed 'North China' to 'East China' everywhere in Section 2. The number of sites in East China is changed to 598 in Table 1. We in addition added in Section 5 in lines 363-366: "The Figure gives normalized mean biases ($NMB$s) relative to ground-based measurements from AERONET and from the $PM_{2.5}$ surface networks (shown as circles) over the North China region (115.5-122° E, 34.5-40.5° N) and South Korea. The North China region is defined to overlap with the domain of the geostationary satellite AOD, and to ensure consistent seasonal variations within its narrow latitude."

References:
Eck, T.F., Holben, B.N., Sinyuk, A., Pinker, R.T., Goloub, P., Chen, H., Chatenet, B., Li, Z., Singh, R.P., Tripathi, S.N. and Reid, J.S., 2010. Climatological aspects of the optical properties of fine/coarse mode aerosol mixtures. Journal of Geophysical Research: Atmospheres, 115(D19).

Li, Z. J. Guo, A. Ding, H. Liao, J. Liu, Y. Sun, T. Wang, H. Xue, H. Zhang, and B. Zhu, 2017: Aerosols and boundary-layer interactions and impact on air quality, Natl. Sci. Rev., 4, 810-833, doi:10.1093/nsr/nwx117.

Chen, J., Yin, J., Zang, L., Zhang, T., and Zhao, M.: Stacking machine learning model for estimating hourly $PM_{2.5}$ in China based on Himawari-8 aerosol optical depth data, Sci. Total Environ., 697, 134021, https://doi.org/10.1016/j.scitotenv.2019.134021, 2019.

**Report #3 from Referee #1**

Line 73: Better to say artificial intelligence algorithms and involve some literature using deep learning models. Also, a typical study focusing on hourly PM estimates using Himawari-8 (https://doi.org/10.5194/acp-21-7863-2021) should be considered.

Changed to "artificial intelligence algorithms". Wei et al. 2021a is cited in line 62 and line 75.

Figure 6: I would like to see the comparison in spatial patterns between satellite and model $PM_{2.5}$ observations in a specific region. Maybe you can use the public CHAP dataset for a simple comparison in Eastern China, available at: https://doi.org/10.5281/zenodo.4660858.

We believe that it would be misleading and detrimental to do this because of our point in the paper that models such as GEOS-Chem need to fix their overestimate of nitrate and the underestimate of coarse PM before attempting to produce a $PM_{2.5}$ product (lines 470-473 in the manuscript) – we don't actually show AOD-inferred $PM_{2.5}$ anywhere. We are now working to improve nitrate and coarse PM in GEOS-Chem, and after we do, we intend to generate a $PM_{2.5}$ product using GEOS-Chem combined with GOCI. Showing a flawed product at this stage would be counterproductive.

**Additional revisions**

1. We updated Jun Meng's affiliation.

2. We changed the color bars in Figure 4, Figure 6, and Figure S7, and the red colors in Figure 7.